Effectiveness of dexmedetomidine as a premedication for pediatric patients undergoing outpatient dental surgery under general anesthesia-systematic review and meta-analysis

Zhang Xiao
Fan Ze
He Danyi
Liu Yang
Shi Xiaotong
Zhang Haopeng zhanghaopeng@fmmu.edu.cn
State Key Laboratory of Oral & Maxillofacial Reconstruction and Regeneration, National Clinical Research Center for Oral Diseases, Shaanxi Engineering Research Center for Dental Materials and Advanced Manufacture, Department of Anesthesiology, School of Stomatology, Fourth Military Medical University, Xi’an, Shaanxi, China
Abu Hasna Amjad
Electronic publication date: 2025 Mar 31
Publication date: 2025
Volume: 13
Electronic Location ID: e19216
Received 2024 Sep 24; Accepted 2025 Mar 5
Copyright: © 2025 Zhang et al.
Copyright year: 2025
Copyright holder: Zhang et al.
License: This is an open access article distributed under the terms of the Creative Commons Attribution License, which permits unrestricted use, distribution, reproduction and adaptation in any medium and for any purpose provided that it is properly attributed. For attribution, the original author(s), title, publication source (PeerJ) and either DOI or URL of the article must be cited.
License URL: https://creativecommons.org/licenses/by/4.0/

Keywords: Child, Sedation, General anesthesia, Emergence delirium

Funding: The authors received no funding for this work.

==============================
Pediatric patients undergoing outpatient dental surgery often exhibit instinctive resistance and fear in face of the unknown, which in turn creates obstacles to subsequent treatment. Appropriate sedation can increase compliance, decrease the incidence of adverse events, and lead to improved treatment outcomes. To evaluate the effectiveness of dexmedetomidine as pre-medication in children undergoing tooth extraction with general anesthesia, we searched databases including the Medline, Embase and Cochrane library for eligible trials. Randomized controlled trials of dexmedetomidine for premedication vs. placebo or midazolam in pediatric patients were included, while trials involving children having dental treatment under local anesthesia were excluded. Two review authors independently participated in the inclusion of trials and assessment of bias. Decisions were made based on group discussion. We finally included seven trials in this review, with a total of 496 pediatric patients. Three of the included trials compared dexmedetomidine with placebo. A meta-analysis indicated that dexmedetomidine was effective for preoperative sedation and for preventing emergence delirium; two of these studies reported no incidences of bradycardia and hypoxemia during study observation period. Four trials compared dexmedetomidine with midazolam, meta-analysis of these four trials gave risk ratios (RR) for successful parental separation, satisfactory mask acceptance and emergence delirium rate of 1.26 (95% CI [0.86–1.84]); 1.07(95% CI [0.94–1.22]); −0.18(95% CI [−0.29 to −0.07]), respectively. Only one study reported complications arising from administration of premedication. Based on the current evidence, we can conclude that the dexmedetomidine appears to be an effective premedication, offering the advantage of reducing the incidence of postoperative delirium compared to midazolam. However, high-quality clinical trials with larger sample sizes are needed to determine the selection of different medication modes and doses, as well as to study perioperative adverse reactions.

Introduction

Children who undergo tooth extraction often experience pain or fear, which is typically accompanied by crying, struggling and anxiety. They can hardly control their emotions and behaviors, and poor treatment compliance became obstacles to subsequent medical procedures (Oriby, 2019; Wright et al., 2007). Moreover, the negative impact of such terrible experiences in childhood can persist into adulthood (Tellez et al., 2015). The judicious use of general anesthesia and preoperative sedatives can be particularly effective in ensuring better patient compliance and preventing adverse effects (Xing et al., 2024). Therefore, comfort measures taken before tooth extraction deserve more attention from anesthesiologist and dental practitioners, and should be considered as standard practice prior to dental treatment.

Dexmedetomidine and midazolam are two sedative drugs commonly applied in clinical practice. Midazolam belongs to classical benzodiazepine, a type of psychoactive drug that exerts relaxing, calming, hypnotic and anterograde amnesia effects by activating the GABA receptor of the ascending reticular activating system (Lourenço-Matharu, Ashley & Furness, 2012). But the application of midazolam may be associated with bronchial hyperresponsiveness and postoperative delirium (Shen et al., 2022; Zarour et al., 2024). Dexmedetomidine is an agonist of α2-adrenergic receptors, expressed in the central nervous system, similar to clonidine, known for its safety and arousal sedation effects (Fink et al., 2022; Qiu et al., 2022). Compared to other commonly used sedatives such as midazolam, propofol, and fentanyl, Dexmedetomidine avoids the risk of respiratory depression and is increasingly widely used in pediatrics (Azimaraghi et al., 2024; Lin et al., 2022; Morse, Cortinez & Anderson, 2021).

Nevertheless, it is still controversial whether dexmedetomidine can supply effective preoperative sedation and analgesia for children undergoing tooth extraction, potentially surpassing midazolam. There is a lack of a systematic summary of studies on this topic. Consequently, we conducted this review to offer anesthesiologist evidence to inform anaesthetic decision for pediatric patients undergoing outpatient dental surgery under general anaethesis.

Materials and Methods

Search strategy

We conducted a systematic search from databases in April 2024, including Embase, PubMed/Medline, and the Cochrane Library using Medical Subject Headings (MeSH) terms and text keywords as follows: ‘Tooth Extraction’, ‘Dental Extraction’, ‘Dental Rehabilitation’, ‘dexmedetomidine’, ‘midazolam’. The search strategies are detailed in Table 1. No language restrictions were imposed. We examined the reference lists of all included studies for additional articles and contacted corresponding authors for any unpublished data. This review was conducted following the guidelines provided by the Cochrane Collaboration and the Preferred Reporting Items for Systematic Reviews and Meta-Analysis (PRISMA) statement. We registered protocol for this study early on PROSPERO on April 3, 2023 (CRD42024531795).

Table 1 Search strategies.

Database	Strategy	Results	
PubMed/Medline	(“Tooth extraction” [Mesh] OR “Extraction*, Tooth” [Title/Abstract] OR “Tooth Extraction*” [Title/Abstract] OR “Dental Extraction*” [Title/Abstract] OR “Dental Rehabilitation*” [Title/Abstract]
) AND (Dexmedetomidine [Title/Abstract] OR “Dexmedetomidine”[Mesh] OR “midazolam”[Mesh] OR midazolam [Title/Abstract]) AND (randomized control trial [Filter])	98	
Embase	((‘Extraction*, Tooth’ OR ‘Tooth Extraction*’ OR ‘Dental Extraction*’ OR ‘Dental Rehabilitation*’) AND (‘Dexmedetomidine’ OR ‘midazolam’)) AND ’randomized controlled trial’/de	103	
Cochrane Library	#1 MeSH descriptor: [Tooth extraction] explode all trees	2,474	
#2 (Extraction*, Tooth):ti,ab,kw OR (Tooth Extraction*):ti,ab,kw OR (Dental Extraction*):ti,ab,kw OR (Dental Rehabilitation*):ti,ab,kw	7,775	
#3 MeSH descriptor: [Dexmedetomidine] explode all trees	2,901	
#4 MeSH descriptor: [midazolam] explode all trees	3,718	
#5 (Dexmedetomidine):ti,ab,kw OR (midazolam):ti,ab,kw	18,522	
#6 #1 OR #2	7,775	
#7 #3 OR #4 OR #5	18,522	
#8 #6 AND #7	195	
Total (as of March 25, 2024):	195	
Note:

*Wildcard character, represent any number of arbitrary characters.

Eligibility criteria

Articles meeting all the following pre-defined criteria were included: 1) randomized controlled trials (RCTs); 2) pediatric patients undergoing tooth extraction with premedication of sedation; 3) comparing the sedative effect of dexmedetomidine with a placebo or midazolam; 4) reporting outcomes of interest, such as the sedation success rate and the incidence of adverse reactions. Exclusion criteria were studies with 1) only local anesthesia; 2) duplicate population cohort; 3) lack of main outcomes. Summarizing study rationale including population, intervention, comparator, outcome, and study design (PICOS) was demonstrated in Table 2.

Table 2 The summarizing study rationale.

Population	Children (<18) undergoing tooth extraction with sedation	
Interventions	Intraoperative dexmedetomidine infusion	
Comparisons	Intraoperative midazolam or placebo control infusion	
Outcomes	Satisfactory Sedation Rate (SSR)
Satisfactory Parental Separation Rate (SPSR)
Mask Acceptance Rate (MAR)
Emergence Delirium Rate (EDR)	
Study design	Only randomized controlled trial	

Outcomes

In this study, we mainly evaluate the sedative effect of premedication using the successful parental separation rate (SPSR) and mask acceptance satisfactory rate (MASR). The sedation score at separation from parents, as determined by the Parental Separation Anxiety Scale (PSAS 1 = easy separation; 2 = whimpers, but easily separated; 3 = cries and cannot be easily reassured, but not clinging to parents; and 4 = crying and clinging to parents), scores of 1 or 2 were considered successful parental separation. The Mask Acceptance Scale (MAS), where one indicates unafraid, cooperative, accepts mask readily, two signifies slight fear of mask but can be easily calmed, three represents moderate fear of the mask and not easily calmed, and four denotes terrified, crying, or struggling, considers scores of 1 or 2 as successful. Additionally, emergence delirium (ED) is assessed using the postoperative emergence delirium scale (PEDS) or pediatric anesthesia emergence delirium scale (PAEDS). A PEDS score of 1/2, or a PAEDS score of 10 or higher, indicates the presence of ED. Postoperative agitation scale (PAS) score of 3/4 indicated the presence of agitation. hemodynamics parameters Adverse events, including postoperative pain, postoperative nausea and vomiting (PONV), hypotension (systolic blood pressure < 70 mmHg), and bradycardia (heart rate < 60 bpm), were taken into account.

Study identification and data extraction

Articles were searched, extracted, and screened independently by two reviewers (Xiao Zhang and Ze Fan). Articles identified as potentially relevant by two reviewers were retained, and duplicates were removed. Discrepancies were resolved through group discussions. Two reviewers independently extracted and assessed the data from included articles. The original quantitative data or qualitative data were selected from charts within the articles, while any data not shown was obtained by contacting the original authors.

Quality assessment

The assessment of RCTs was conducted using Cochrane’s tool (Cochrane Library’s Review Manager software, RevMan version 5.4.1) for determining the risk of bias and methodological quality. Each study was assessed in five domains: selection, performance, detection, attrition, and reporting, with each domain scored as: high, low, or unclear risk of bias. The risk of bias assessments were performed by two independent authors (Danyi He and Xiaotong Shi). Any conflicts arising between the authors were resolved through group discussions, where the reasons for discrepancies were presented, and consensus was ultimately achieved.

Statistical analysis

We presented binary outcomes (e.g., SPSR) as risk ratios (RRs) with their associated 95% confidence intervals (CIs). Mean differences and their 95% CIs were used for continuous outcomes. We calculated the I2 statistic, which indicates the percentage of heterogeneity associated with total variation and performed Cochran’s test for heterogeneity for each meta-analysis (Migliavaca et al., 2022; Wang, DelRocco & Lin, 2024). A random-effect model was used when I2 > 50%. We proposed conducting subgroup analyses according to drug dosage and routes of administration medication if data were available. Unless otherwise specified, a p-value of less than 0.05 (two-sided) was considered statistically significant.

Results

Characteristics of included studies

A total of 236 citations were identified after de-duplication. Following the screening titles and abstracts, 131 records were excluded. Of these, 149 trials were related to adults, 46 trials were protocols, 22 trials lacked a control group or relevant data, and 1 was an animal trial. Overall, 16 RCTs were chosen for full-text screening. Out of these, nine studies were excluded due to insufficient data (Done et al., 2016; Hao et al., 2017; Moreira et al., 2013; Oriby, 2019; Roelofse et al., 2004; Shama et al., 2023; Van der Bijl, Roelofse & Stander, 1991; Wang et al., 2020; Xiaohua, Yanzhong & Li, 2015) and duplicate record (Sathyamoorthy et al., 2016). Finally, seven studies consisting of 496 pediatric patients were included in this review. There were no significant disagreements among the review authors during the screening process. The study selection flowchart is presented in Fig. 1. All studies adhered to parallel group RCT designs. The sample sizes in seven studies ranged from 41 to 100 participants. Only two studies did not report the sample size calculation process (Gao, Liu & Yang, 2018; Mountain et al., 2011). All the included RCTs were single-center studies. Three studies (42.9%) were conducted in China, (Gao, Liu & Yang, 2018; He et al., 2023; Wang et al., 2020) and two studies (28.6%) were completed in United States, (Mountain et al., 2011; Sathyamoorthy et al., 2019) with the remaining two studies were performed in Turkey (Keles & Kocaturk, 2017) and Saudi Arabia (Sheta, Al-Sarheed & Abdelhalim, 2013). The population included in this study were all children but each study included a different age group ranging from 1 to 18 years old. Four of the seven included studies compared the sedative effect of dexmedetomidine and midazolam before general anesthesia, (Mountain et al., 2011; Sathyamoorthy et al., 2019; Sheta, Al-Sarheed & Abdelhalim, 2013; Wang et al., 2020) while the remaining three studies evaluated the sedative effect of dexmedetomidine compared with a placebo control group (Gao, Liu & Yang, 2018; He et al., 2023; Keles & Kocaturk, 2017). Table 3 provides details of the characteristics and summaries of primary outcomes for all included studies.

Figure 1 Study selection flowchart.

Table 3 Main characteristics of included studies.

Study (year)	Study period	Study design	Country	Sample	Age
(years range)	ASA status	Dexmedetomidine
(dose)	Control (dose)	Outcomes	
Wang et al. (2020)	2019.01–2019.08	RCT (parallel)	China	60	3~6	I	2 ug/kg intranasal	0.5 mg/kg
midazolam oral	RSS PSAS MAS EDS
Hemodynamic index	
Mountain et al. (2011)	2006.05–2007.06	RCT (parallel)	United States	41	1~6	I	4 ug/kg oral	0.5 mg/kg
midazolam oral	PSAS MAS EDS Hemodynamic index	
Sheta, Al-Sarheed & Abdelhalim (2013)		RCT (parallel)	Saudi Arabia	72	3~6	I & II	1 ug/kg intranasal	0.2 mg/kg
midazolam intranasal	PSAS MAS EDS CHEOPS PONV Hemodynamic index	
Sathyamoorthy et al. (2016)	2014.09–2016.09	RCT (parallel)	American	73	5~18		2 ug/kg intranasal	0.5 mg/kg
midazolam oral	PSAS MAS EDS Hemodynamic index	
Gao, Liu & Yang (2018)		RCT (parallel)	China	60	2~9	I & II	2 ug/kg intranasal	Saline intranasal	RSS PSAS MAS EDS Hemodynamic index	
He et al. (2023)	2021.02–2022.04	RCT (parallel)	China	90	3~7	I & II	1/2 ug/kg intranasal	Saline intranasal	RSS PSAS MAS EDS Hemodynamic index	
Keles & Kocaturk (2017)	2015.12–2016.07	RCT (parallel)	Turkey	100	2~6	I	1 ug/kg oral	Juice oral	RSS PSAS MAS EDS Hemodynamic index	

The included studies described as employing randomization did not report the specific methods of sequence generation and allocation concealment, leading us to typically considered them as at unclear risk of bias. All studies followed the principle of blinding including single blind (Sathyamoorthy et al., 2019) and double blind methods (Gao, Liu & Yang, 2018; He et al., 2023; Keles & Kocaturk, 2017; Mountain et al., 2011; Sheta, Al-Sarheed & Abdelhalim, 2013; Wang et al., 2020). Three studies that did not describe blinding of outcome assessment were deemed to have an unclear risk of bias. Only one study was assessed as having a high risk due to unspecified baseline data (Mountain et al., 2011). The results of the bias analysis are shown in Fig. 2.

Figure 2 Quality assessment of included studies.

Comparison 1: dexmedetomidine vs placebo

Satisfactory parental separation rate

PSAS were recorded in all three studies, (Gao, Liu & Yang, 2018; He et al., 2023; Keles & Kocaturk, 2017) and the results showed that patients in the dexmedetomidine group experienced higher satisfactory parental separation rate (SPSR) (RR 1.54; 95% CI [1.24–1.91]; p < 0.0001; I2 = 0%; Fig. 3A).

Figure 3 Comparison 1. Dexmedetomidine vs. placebo (Gao, Liu & Yang, 2018; He et al., 2023; Keles & Kocaturk, 2017).

Mask acceptance rate

MAS were recorded in all three studies, the results indicated that patients in the dexmedetomidine group would have mask acceptance rate (MAR) (RR 1.76; 95% CI [1.40–2.20]; p < 0.00001; I2 = 0%; Fig. 3B).

Emergence delirium rate

PAEDS were recorded in two studies, (He et al., 2023; Keles & Kocaturk, 2017) where it was found that the dexmedetomidine could significantly reduce the incidence of postoperative delirium in children (RR 0.36; 95% CI [0.22–0.57]; p < 0.0001; I2 = 0%; Fig. 3C).

Other outcomes

All studies compared hemodynamic data between the two groups before and after sedation and during surgery, and did not show any statistical difference. PAS were recorded in one study, (Gao, Liu & Yang, 2018) and there was no significant difference among groups with low incidence of postoperative agitation.

Comparison 2: dexmedetomidine vs midazolam

Successful parental separation rate

PSAS recorded in two studies (Mountain et al., 2011; Wang et al., 2020) and sedation scores at separation from parent in two studies (Sathyamoorthy et al., 2019; Sheta, Al-Sarheed & Abdelhalim, 2013) were used to evaluate SPSR. SPSR was obviously higher in the dexmedetomidine group compared with the midazolam group using fixed model (RR 1.29; 95% CI [1.10–1.50]; p = 0.001; I2 = 88%; Fig. 4A). The significant heterogeneity may stem from age difference between groups, we then use the random effects model to analyze the above results, which indicates that the difference is not statistically significant (RR 1.26; 95% CI [0.86–1.84]; p = 0.24; Fig. 4A).

Figure 4 Comparison 2. Dexmedetomidine vs. midazolam (Wang et al., 2020; Mountain et al., 2011; Sheta, Al-Sarheed & Abdelhalim, 2013; Sathyamoorthy et al., 2019).

Mask acceptance rate

MAS were recorded in all four studies, the difference in MAR between dexmedetomidine and midazolam groups was not statistically significant (RR 1.07; 95% CI [0.94–1.22]; p = 0.29; I2 = 38%; Fig. 4B).

Emergence delirium rate

PEDS were recorded in one study, (Sheta, Al-Sarheed & Abdelhalim, 2013) and PAEDS were recorded in two studies (Mountain et al., 2011; Wang et al., 2020). EDR was distinctly higher in the midazolam group compared with the dexmedetomidine group using fixed model (RR 0.37; 95% CI [0.17–0.80]; p = 0.01; I2 = 0%; Fig. 4C).

Other outcomes

Two studies compared hemodynamic data (mean HR, SpO2, and RR) between the two groups before and after sedation until anesthesia induction (Sathyamoorthy et al., 2019; Wang et al., 2020). Two studies recorded the above parameters throughout the trial, (Mountain et al., 2011; Sheta, Al-Sarheed & Abdelhalim, 2013) and did not show any statistical difference. One study analyzed the incidence of various adverse effects in two groups, (Sheta, Al-Sarheed & Abdelhalim, 2013) indicating that children in the intranasal dexmedetomidine group suffered less postoperative pain, required fewer rescue analgesics, and had a lower incidence of shivering (p < 0.05). Side effects of nasal irritation occurred only in the midazolam group. However, there was no significant difference in the incidence of postoperative nausea and vomiting (PONV).

Discussion

For children, particularly the young ones, the administration of preoperative sedative drugs is usually necessary to enhance cooperation and prevent a series of postoperative adverse consequences caused by preoperative anxiety and fear (Wright et al., 2007). Dexmedetomidine and midazolam are commonly used sedative medications in clinical practice (Bd et al., 2023; Cai et al., 2021; de Rover et al., 2022; Min et al., 2016). But for outpatient pediatric tooth extractions, there is currently no clear consensus on the selection of preoperative sedatives. Therefore, we conducted this study to provide evidence to assist physicians in clinical decision-making by physicians.

This review consists of seven studies and 496 pediatric patients, comparing the sedative effects of dexmedetomidine premedication with those of midazolam or placebo in pediatric patients undergoing tooth extraction. The included studies were all RCT’s published between 2006 and 2022. The baseline data, which includes age, gender, weight, anesthesia, and operation time of pediatric patients, were relatively balanced between the two groups.

Midazolam is a classic sedative-hypnotic drug with anxiolytic and retrograde amnesic properties commonly used in clinical practice. The sedative effect can be evaluated by the rate of successful parental separation and the rate of mask acceptance. In the meta-analysis of the successful parental separation rate, the results from our fixed-effect model showed that dexmedetomidine was more effective than midazolam. However, due to significant heterogeneity, the result was deemed unreliable. When switching to a random-effects model, the differences between dexmedetomidine and midazolam were found to be statistically insignificant. Similarly, there was no statistical difference in the meta-analysis of mask acceptance rates, although dexmedetomidine exhibited a greater sedative effect than midazolam. Study in 2019 included older children, ranging in age from 5 to 18, with an average age of 7 years, (Sathyamoorthy et al., 2019) which was larger than the other studies. The results of the meta-analysis, after excluding this trial, were consistent with those obtained before exclusion. Thus, we can conclude that dexmedetomidine has a similar preoperative sedation effect to midazolam. Moreover, in the meta-analysis of ED, our pooled results clearly suggested that compared with the midazolam group, the incidence of ED was lower in the dexmedetomidine group (p = 0.01) with low heterogeneity. However, these trials do not adequately control for baseline anxiety levels, which may obscure the true effectiveness of premedications. And these differences can be relatively balanced and comparable through strict randomization and blinding, which we should put strong emphasis on. The other undesirable postoperative side effects of general anesthesia, such as nausea and vomiting, negative behavior change, are more common in pediatric patients with dental surgery (Dobbeleir et al., 2018; Shama et al., 2023). We got only one study comparing the postoperative adverse effects, the results showed that dexmedetomidine caused less nasal irritation than midazolam, which may be related to the pharmacological properties of midazolam itself. There is a study suggesting that the incidence of perioperative respiratory adverse events (PRAE) increases with prenasal administration of midazolam in children undergoing tonsillectomy and adenoidectomy, compared to dexmedetomidine (Ayerza Casas, Ayerza Casas & Crespo Escudero, 2017; Shen et al., 2022). The evidence of the impact of dexmedetomidine on reducing postoperative adverse effects in child varies significantly, which necessitates further strengthening (Azimaraghi et al., 2024; Lee-Archer et al., 2020; Zhou et al., 2023).

However, it is important to note that in the meta-analysis of comparing dexmedetomidine to midazolam, there were differences in the modes of administration and drug dosages. These variations in administration may lead to failure in blinding. After our group discussion, we concluded that various modes of administration did not affect study outcomes, as drug administration and data collection were conducted by different investigators. In addition, the intranasal dose of dexmedetomidine is 1 to 2 ug/kg, the oral dose is 4 ug/kg. For midazolam, the intranasal dose is 0.2 mg/kg, the oral dose is 0.5 mg/kg. We hoped that subgroup analyses could be performed according to dose and mode of administration, but the number of included studies was too small to support. Indeed, RCTs on the use of sedatives for premedication before tooth extraction in pediatric patients are relatively few in number. Most of these are RCTs with small sample sizes and lacking follow-up, varying in quality, and the postoperative adverse reactions are not evaluated as the primary outcome. Therefore, RCTs with larger sample sizes and longer follow-up periods to assess postoperative adverse reactions are needed to instruct the use of preoperative sedative drugs.

Conclusion

Based on the current evidence, we can conclude that intranasal dexmedetomidine as premedication for children undergoing tooth extraction may have a favorable sedative effect. Children experience less anxiety and fear when separated from their parents, are more cooperative during anesthesia induction, and are less likely to suffer from postoperative delirium. In addition, the sedative effect of dexmedetomidine is comparable to that of midazolam, being less irritating and having a lower incidence of postoperative delirium. We believe that dexmedetomidine could be a superior optiion when sedation is required. However, RCTs with larger sample sizes and longer follow-up periods are necessary to further evaluate the efficacy and safety of dexmedetomidine for sedation during tooth extraction in pediatric patients.

Supplemental Information

Supplemental Information 1 PRISMA checklist.

Supplemental Information 2 Rationale and Contribution.

Thanks to Dr. Huan He from School of Stomatology Fudan University for providing the original data.

Abbreviations

RCT Randomized controlled trials

MESH Medical Subject Heading

SPSR Successful parental separation rate

MASR Mask acceptance satisfactory rate

ED Emergence delirium

PEDS Postoperative emergence delirium scale

PAEDS Pediatric anesthesia emergence delirium scale

PAS Postoperative agitation scale

PONV Postoperative nausea and vomiting

Additional Information and Declarations

Competing Interests

The authors declare that they have no competing interests.

Author Contributions

Xiao Zhang conceived and designed the experiments, performed the experiments, analyzed the data, authored or reviewed drafts of the article, and approved the final draft.

Ze Fan performed the experiments, prepared figures and/or tables, and approved the final draft.

Danyi He analyzed the data, prepared figures and/or tables, and approved the final draft.

Yang Liu performed the experiments, prepared figures and/or tables, and approved the final draft.

Xiaotong Shi analyzed the data, prepared figures and/or tables, and approved the final draft.

Haopeng Zhang conceived and designed the experiments, authored or reviewed drafts of the article, and approved the final draft.

Data Availability

The following information was supplied regarding data availability:

The search strategy is available in Table 1. This is a systematic review/meta-analysis.

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
