# Peer review of "Effectiveness of dexmedetomidine as a premedication for pediatric patients undergoing outpatient dental surgery under general anesthesia-systematic review and meta-analysis"

_PeerJ, doi:10.7717/peerj.19216_

## Round 0.1 · original submission · Major Revisions

Address all the comments from both reviewers (both of whom have requested major revisions)

·

Basic reporting

The paper on the "Sedative and Adverse Effect of Dexmedetomidine for Premedication in Pediatric Outpatient Surgery of Stomatology – Systematic Review and Meta-Analysis" has several areas that could be improved:

Sample Size and Study Selection:

The meta-analysis includes only seven studies with a total of 496 patients, which is a relatively small sample for robust conclusions. This limits the statistical power, and it would benefit from the inclusion of additional studies if available.
The study could address why some relevant trials were excluded, especially if studies with local anesthesia were omitted.

Heterogeneity in Study Parameters:

There is significant variability in the age range, dosage, and method of administration (intranasal, oral) across the included studies. This heterogeneity may affect the reliability of pooled estimates, particularly for outcomes like sedation success rate and emergence delirium. Subgroup analyses by administration method or dosage could provide more nuanced insights.

Limited Outcome Measures and Follow-Up:

Only one study reported postoperative adverse events, which limits the analysis of long-term safety and tolerability of dexmedetomidine. Including more data on postoperative adverse effects and behavioral outcomes would strengthen the conclusions about its safety profile.
Risk of Bias Assessment:

The paper notes that the randomization methods in most studies were unclear, and blinding was inconsistently reported. This could introduce bias, impacting the validity of the findings. Clarifying these aspects would make the study more reliable.



Discussion of Practical Implications:

While the study concludes that dexmedetomidine shows a comparable sedative effect to midazolam, further discussion on the practical implications—such as cost, ease of administration, and caregiver preferences—would improve the clinical relevance of the paper.
Addressing these issues would improve the clarity, reliability, and applicability of the study findings.

Experimental design

Lack of Comparative Analysis with Other Sedatives:

While the study compares dexmedetomidine with midazolam, it could enhance relevance by also comparing it to other sedatives used in pediatric dentistry, such as ketamine, to give a broader perspective on efficacy and safety

Validity of the findings

Statistical Analysis and Heterogeneity:

The high heterogeneity (I² values) reported in some outcomes (like parental separation success rate) suggests potential issues with study compatibility. Employing meta-regression or sensitivity analysis to explore the sources of heterogeneity could add robustness to the conclusions

·

Basic reporting

The English usage needs correction throughout. Please consider asking a colleague to assist with a revision. There is also some repetition of information which could be reduced.

References are adequate, as is the presentation of the paper including the Figures and Tables

Experimental design

The research question is well defined. However, throughout the paper sedation and anxiolytic qualities and to an extent analgesic properties are used inter changably. It is my understnading that dexmedetomidine reduces anxiety but is not sedative unlike midazolam. This confusion should be clarified througout the document.

Validity of the findings

The findings are valid within the confinds of the available data.
Many trials of premedications do not control for anxiety in their sample, meaning that true differences in effectiveness of premedications is not identified becasue of the cohort of non-anxious patients.

Additional comments

1.A more appropriate title would be: " Effectiveness of dexmedetomidine as a premedication for pediatric patients undergoing outpatient dental surgery under general anaethesis- systematic review and meta-analysis" or similar.
Abstract
2. L19 It is not all children only some.
3. L23 an example of "analgesic" being used rather than pre-medication for anxiety.
4. L38 The Conclusion should be reworded and "sure" removed replaced with "dexmedetomine appears and effective premedication".
Intoduction
5. L44 Reword to make it clear you are talkingabout treatment under GA
6. L58 Midazolam has other potential disadvantages such as emergence delium, excitement rather than sedation.
7. L66 Replace "hint" with "evidence to inform anaesthetic decision".
Materials and methods
8. L99 It is not clear when considering post-operative pain if local anaesthetic had been delivered and considered in the analysis.
Discussion
9. L203 Re-phrase sentence to "The included studies were all RCT's published between 2006-2022."
10. The discussion would benefit from text on how future trials could be better designed and reported. ie ensuring only anxious children are included. Using other outcome measures such as M-YPAS
Conclusion
11. L251 I do not disagree that dexmedetomidine may well be "a better choice" of premedication. However, based on the results of this review and the literature this is too strong a statement to make at this point.

I could not check the references because of the formating appears to have corrupted

---

## Round 0.2 · accepted · Accept

Dear authors,

We are pleased to inform you that your manuscript has been accepted for publication in PeerJ after a thorough peer-review process.

The reviewers and editorial board found your work to be a valuable contribution to the field and we appreciate your efforts in addressing the comments provided during the review process.

Congratulations on this achievement! We appreciate your contribution and look forward to sharing your work with the scientific community.

·

Basic reporting

The authors have addressed the points raised.

One smal point is "concludes" is spelt "concluds" at line 554

Experimental design

No comment

Validity of the findings

No comment

Additional comments

No comment